# Comparative Risk of Acute Kidney Injury Following Concurrent Administration of Vancomycin with Piperacillin/Tazobactam or Meropenem: A Systematic Review and Meta-Analysis of Observational Studies

**DOI:** 10.3390/antibiotics11040526

**Published:** 2022-04-14

**Authors:** Abdulmajeed M. Alshehri, Mohammed Y. Alzahrani, Mohammed A. Abujamal, Mariam H. Abdalla, Shuroug A. Alowais, Osamah M. Alfayez, Majed S. Alyami, Abdulaali R. Almutairi, Omar A. Almohammed

**Affiliations:** 1Department of Pharmacy Practice, College of Pharmacy, King Saud bin Abdulaziz University for Health Sciences, Riyadh 14611, Saudi Arabia; abdulmajeed483@gmail.com (A.M.A.); alzahranim019@gmail.com (M.Y.A.); ahmedmo@ksau-hs.edu.sa (M.A.A.); abdallama@ksau-hs.edu.sa (M.H.A.); owaiss@ksau-hs.edu.sa (S.A.A.); yamim@ksau-hs.edu.sa (M.S.A.); 2King Abdulaziz Medical City, National Guard Health Affairs, Riyadh 11426, Saudi Arabia; 3King Abdullah International Medical Research Center, Riyadh 11481, Saudi Arabia; 4Department of Pharmacy Practice, College of Pharmacy, Qassim University, Buraydah 51452, Saudi Arabia; oalfayez@qu.edu.sa; 5Drug Sector, Saudi Food and Drug Authority, Riyadh 13513, Saudi Arabia; armutairi@sfda.gov.sa; 6Department of Clinical Pharmacy, College of Pharmacy, King Saud University, Riyadh 12372, Saudi Arabia; 7Pharmacoeconomics Research Unit, College of Pharmacy, King Saud University, Riyadh 12372, Saudi Arabia

**Keywords:** acute kidney injury, vancomycin, piperacillin-tazobactam, meropenem, nephrotoxicity

## Abstract

The study aims to comparatively assess the nephrotoxicity of vancomycin when combined with piperacillin-tazobactam (V + PT) or meropenem (V + M) in adult patients hospitalized in general wards or intensive care units. We searched MEDLINE, Google Scholar, and Web of Science for observational studies evaluating incidences of AKI in adult patients receiving V + PT or V + M for at least 48 h in general wards or intensive care units. The primary outcome was AKI events, while the secondary outcomes were hospital length of stay, need for renal replacement therapy (RRT), and mortality events. The odds ratio (OR), or mean difference for the hospital length of stay, with a corresponding 95% confidence interval (CI) from the inverse variance weighting random-effects model were estimated for the risk of AKI, RRT, and mortality. Of the 112 studies identified, twelve observational studies were included in this meta-analysis with a total of 14,511 patients. The odds of having AKI were significantly higher in patients receiving V + PT compared with V + M (OR = 2.31; 95%CI 1.69–3.15). There were no differences between V + PT and V + M in the hospital length of stay, RRT, or mortality outcomes. Thus, clinicians should be vigilant while using V + PT, especially in patients who are at high risk of AKI.

## 1. Introduction

Vancomycin has good activity against Gram-positive pathogens and can be used for multiple types of infection. However, it is known for its risk of causing acute kidney injury (AKI) by inducing acute interstitial nephritis and/or acute tubular necrosis [1]. The risk of vancomycin-induced AKI has been widely reported and ranges between 5 and 7% [1]. Incidences of AKI are associated with increased health-related costs, prolonged length of hospitalization, and higher mortality and morbidity rate, especially among hospitalized patients [2,3,4,5,6,7]. Several risk factors for vancomycin-induced AKI are known, for example, the dose of vancomycin, duration of therapy, plasma level, age of patient, comorbidities, concurrent use of other nephrotoxic agents, and admission to the intensive care unit [8,9,10,11,12,13,14].

The combination of vancomycin/piperacillin-tazobactam (V + PT) is one of the most commonly used broad-coverage antibiotics in hospitals [12]. It provides activity against anaerobic bacteria, Gram-negative pathogens, including Pseudomonas aeruginosa, and Gram-positive pathogens, such as methicillin-resistant Staphylococcus aureus (MRSA). However, several studies have assessed the nephrotoxic effect of V + PT in patients treated in critical and non-critical care settings and found it to be associated with an increased risk of developing AKI [13,14,15,16,17,18,19,20,21,22,23,24,25,26,27,28,29,30,31]. Consequently, the use of other antipseudomonal beta-lactam antibiotics, such as cefepime and meropenem, was suggested alongside of vancomycin as an alternative for V + PT to avoid or minimize the risk of AKI.

Several studies have investigated the difference in the nephrotoxic effect for vancomycin when it is combined with either PT or other antipseudomonal beta-lactam antibiotics [21,28,30,31,32,33,34,35,36,37,38,39]. Moreover, conflicting results were reported regarding the incidences of AKI with V + PT compared to vancomycin/meropenem (V + M). Therefore, this meta-analysis seeks to evaluate the nephrotoxic effect of V + PT in comparison to V + M.

## 2. Materials and Methods

### 2.1. Data Source and Search Strategy

A systematic search was conducted using MEDLINE, Google Scholar, and Web of Science to identify observational studies evaluating incidences of AKI in adult patients receiving V + PT or V + M between January 2017 and November 2021. Search terms included piperacillin-tazobactam, vancomycin, meropenem, acute kidney injury, and nephrotoxicity. Moreover, bibliographies of recent reviews and meta-analyses were manually searched to identify further studies.

### 2.2. Study Selection

We included observational studies published in peer-reviewed journals and reported the incidences of AKI in patients receiving V + PT versus V + M for at least 48 h (Appendix A). Studies published in a non-English language or published as an abstract were excluded. Two investigators (MAA and MSA) combined the citations generated from searching the databases and removed the duplicates, then screened the title and abstracts independently. The assessment of the full text articles for inclusion was completed by two independent investigators (MAA and MSA) and verified by a third investigator (MHA).

### 2.3. Data Extraction, Risk of Bias Assessment, and Statistical Analysis

For each study, the first author’s name, study design, sample size, study location, year of publication, clinical setting (intensive care units (ICU) or non-ICU), and AKI definition used were extracted. We also extracted data on the primary outcome of interest: incidences of AKI, as well as the secondary outcomes (length of stay (LOS) in hospital, renal replacement therapy (RRT), and mortality) and the factors included in the adjusted analysis. The risk of bias assessment was conducted for each study using the Newcastle–Ottawa Scale (NOS) for assessing the quality of non-randomized studies in meta-analysis by two independent investigators (MAA and MSA) [40]. The odds ratio (OR) with the corresponding 95% confidence interval (CI) from the inverse variance weighting random-effects model was estimated for the risk of AKI, RRT, and mortality. The mean difference was estimated using the inverse variance weighting random-effects model to estimate the difference in the LOS in hospitals between V + PT and V + M users. Heterogeneity was assessed using I^2^ statistics. We also conducted a subgroup analysis based on the clinical setting (ICU, non-ICU, or both). Moreover, we conducted a sensitivity analysis for studies that involve adjustment for potential confounders to estimate the adjusted odds ratio (aOR) for AKI. All the analyses were conducted using the RevMan 5.3 software (The Nordic Cochrane Centre, Copenhagen, Denmark). This meta-analysis was prepared following the preferred reporting system for meta-analysis of observational studies (MOOSE) [41].

## 3. Results

### 3.1. Study Characteristics

A total of 112 observational studies were initially identified in the literature search, twelve of which were included in this meta-analysis, with a total of 14,511 patients [21,28,30,31,32,33,34,35,36,37,38,39]. One-hundred studies were excluded based on population, measured outcomes, and relevancy to the objective of the meta-analysis (Figure 1). The sample size for the included studies ranged between 76 and 10,236 patients (Table 1). Of the included studies, eleven were retrospective in nature and one was prospective. Four studies were conducted including both critically and non-critically ill patient populations, while five studies were conducted including non-critically ill patients and three included critically ill patients from the ICU only. Six studies used the Kidney Disease Improving Global Outcomes (KDIGO) criteria for AKI definition [42]; two used the Acute Kidney Injury Network (AKIN) criteria [43]; one used the Risk, Injury, Failure, Loss of kidney function, and End-stage kidney disease (RIFLE) criteria [44], whereas three studies used pre-specified criteria, defined as an increase in serum creatinine by 0.5 mg/dL or 50% above the baseline. The NOS score for all included studies was 7–9 (Appendix A).

### 3.2. Outcomes from the Main Analysis

#### 3.2.1. AKI

The pooled analysis for studies reporting the risk of AKI indicated that the odds of developing AKI were significantly higher in patients who received V + PT versus those who received V + M (OR = 2.31; 95%CI 1.69–3.15, I^2^ = 59%; Figure 2). The pooled analysis of the aORs also showed increased odds of AKI following the use of V + PT versus V + M (aOR 2.72; 95%CI 1.82–4.07, I^2^ = 55%; Appendix A).

#### 3.2.2. LOS in Hospital

Although there was a mean difference of approximately half a day between the two groups in favor of the V + PT group, this difference between the groups did not reach statistical significance (MD = −0.48 day; 95%CI −2.00–1.04, I^2^ = 73%; Figure 3).

#### 3.2.3. Renal Replacement Therapy (RRT)

A total of 14 patients in the V + PT group required RRT compared with 6 patients in the V + M group. However, this difference between the two groups was not statistically significant (OR = 1.15; 95%CI 0.40–3.27, I^2^ = 0%; Figure 4).

#### 3.2.4. Mortality

The rate of mortality was 11.6% in the V + PT group compared with 15.0% in the V + M group, but this difference was not statistically significant (OR = 0.76; 95%CI 0.46–1.24, I^2^ = 62%; Figure 5).

### 3.3. Outcomes from the Subgroup Analyses Based on Clinical Setting

#### 3.3.1. AKI

The risk of AKI was statistically higher in ICU and non-ICU settings for V + PT versus V + M treated patients (OR = 1.97; 95%CI 1.39–2.79, I^2^ = 43%, and OR = 2.78; 95%CI 2.12–3.64, I^2^ = 0%, respectively; Appendix A). In addition, four studies combined data for patients from both clinical settings. The pooled analysis of these studies showed no statistical difference in the risk of AKI between V + PT and V + M with significant heterogeneity (OR = 1.81; 95%CI 0.57–5.68, I^2^ = 80%; Appendix A). After restricting the analysis to studies that reported an adjusted analysis for potential confounders, the risk of AKI in the ICU setting was not significant (aOR = 2.05; 95%CI 0.93–4.52, I^2^ = 63%; Appendix A). Only one study presented a significant increase in the odds of AKI among V + PT versus V + M users in patients treated in ICU and non-ICU settings with (aOR = 7.03; 95%CI 1.97–25.09; Appendix A).

#### 3.3.2. LOS in Hospital

The LOS was not statistically different between V + PT and V + M users based on the clinical setting (Appendix A).

#### 3.3.3. Renal Replacement Therapy (RRT)

The subgroup analysis based on the clinical setting did not show a difference in the risk of RRT between V + PT users and V + M users (Appendix A).

#### 3.3.4. Mortality

The subgroup analysis included only two studies in the non-ICU setting, and their pooled analysis did not show a difference in mortality between V + PT and V + M users (Appendix A). There was only one study in the ICU setting subgroup and one study in the combined ICU and non-ICU setting subgroup. These studies showed a reduction in the risk of mortality among V + PT users compared to V + M users (Appendix A).

## 4. Discussion

This meta-analysis sought to evaluate the nephrotoxic effect when using vancomycin in combination with either piperacillin/tazobactam or meropenem. The use of vancomycin combined with piperacillin/tazobactam was associated with a higher risk of AKI compared to the use of vancomycin combined with meropenem. However, no difference between the two combinations was observed in the hospital LOS, need for RRT, or mortality. In this meta-analysis, 11 of the 12 included studies showed higher rates of AKI when using V + PT compared to V + M. Although the study by Tookhi et al. reported a contrasting result, higher odds of AKI with V + M compared to V + PT [38], this deviation could be due to the higher number of critically ill patients in the V + M compared to the V + PT group. However, this difference was not statistically significant.

The findings from the overall analyses of the risk of AKI were consistent with those from previous meta-analyses that compared the use of V + PT to vancomycin combined with other beta-lactams; in fact, patients on V + PT had a higher risk of AKI compared to patients on other combinations [45,46]. The meta-analysis of Chen et al. included eight observational studies comparing the risk of AKI in patients receiving V + PT versus patients on vancomycin in combination with either cefepime, meropenem, or cefepime with tobramycin. They found that the use of V + PT was associated with an increase in the risk of AKI compared to the use of vancomycin in combination with meropenem or cefepime [45]. In addition, Giuliano et al. analyzed the results of fifteen observational studies assessing the risk of AKI in patients receiving V + PT versus vancomycin monotherapy or in combination with other antibiotics. Their findings were also consistent with our findings, as the risk of AKI associated with the use of the V + PT combination was significantly higher compared to the use of vancomycin monotherapy or in combination with other antibiotics [46].

Several efforts and recommendations have been proposed in practice to prevent or minimize the risk of AKI while using vancomycin. Examples of preventive measures include hydration, avoiding the concomitant use of other nephrotoxic medications such as NSAIDs, and therapeutic drug monitoring to assure the appropriateness of the vancomycin dose [47]. The result from this study suggests the need for vigilant assessment before using V + PT when possible, to help reduce the risk of AKI in high-risk patients. However, the study does not suggest that using meropenem can result in a lower risk of AKI compared to other beta-lactam antibiotics, such as cefepime; thus, further studies are needed to answer this question.

The main limitation of this meta-analysis is the inclusion of several studies with a heterogeneous population. However, several sub-analyses were conducted to drive the right conclusion. In addition, there were variations in the AKI definition across the included studies. However, all criteria used for defining AKI are commonly used in practice and research; thus, the variation may not affect this study’s conclusions. Although most of the evaluated studies were retrospective, the combined analysis of these studies was justified by the lack of randomized clinical trials evaluating the impact of different combinations on the risk of developing AKI and the fact that retrospective observational studies may provide the best available evidence for practice and data for future research. Nevertheless, our results should be interpreted carefully, and a reasonable assessment of these results should be considered.

## 5. Conclusions

The accumulating evidence suggests that V + PT is associated with an increased risk of AKI compared to vancomycin alone or in combination with other beta-lactam antibiotics [48]. This meta-analysis revealed that the V + PT combination is associated with a greater risk of developing AKI than is V + M. Thus, clinicians should be vigilant when using V + PT, especially in patients with a high risk of AKI.

## Figures and Tables

**Figure 1 antibiotics-11-00526-f001:**
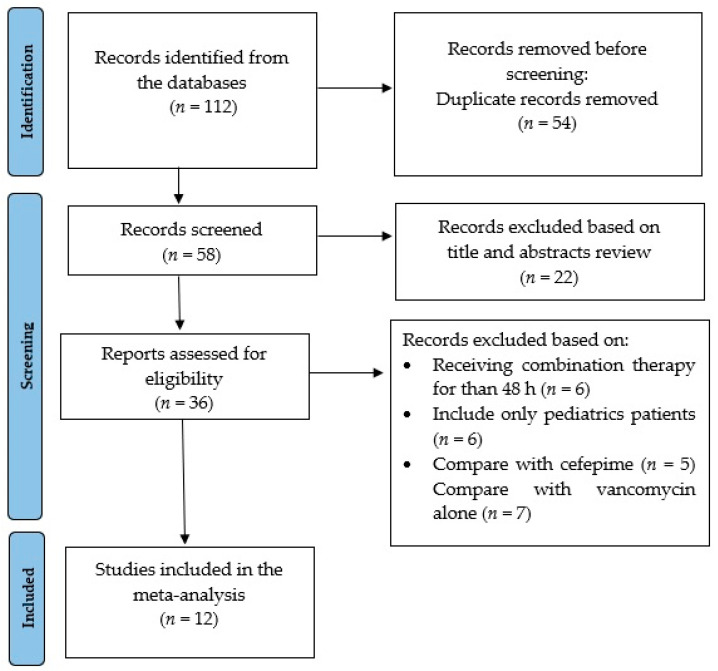
Flow diagram for selection of studies included in the meta-analysis.

**Figure 2 antibiotics-11-00526-f002:**
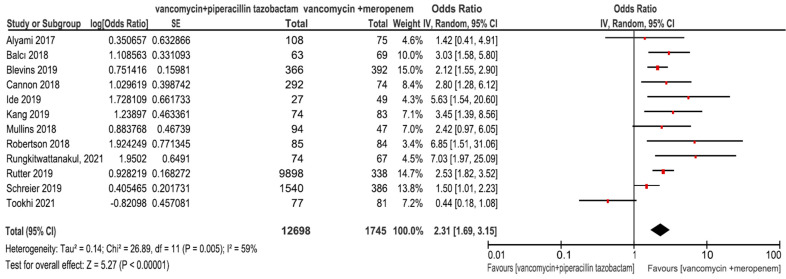
Acute kidney injury (AKI).

**Figure 3 antibiotics-11-00526-f003:**
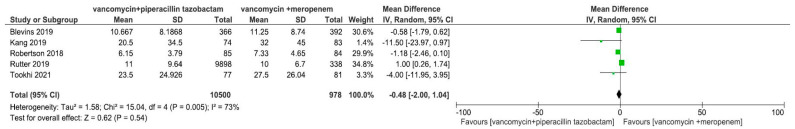
Length of stay (LOS) in hospital.

**Figure 4 antibiotics-11-00526-f004:**
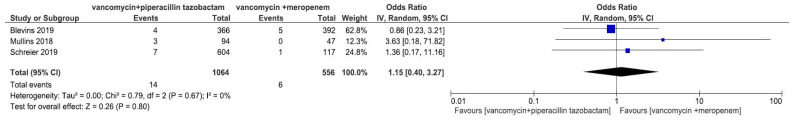
Renal replacement therapy (RRT).

**Figure 5 antibiotics-11-00526-f005:**
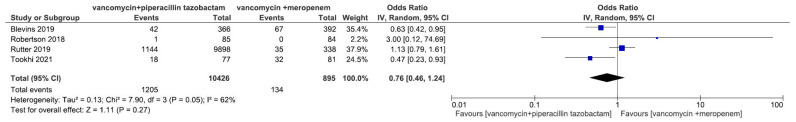
Mortality.

**Table 1 antibiotics-11-00526-t001:** Studies that were included in the systematic review and meta-analysis.

Study	Year	Type	Country	AKI Definition	Proportion of Critically Ill Patients	Patients with AKI History or CKD	Sample Size	Incidence		*p*-Value
								V + PT	V + M	
**Studies including critically and non-critically ill patients**										
Alyami et al.	2017	Retrospective cohort	USA	KDIGO	17.5%	Excluded	183	8/108 (7.4%)	4/75 (5.3%)	0.4
Cannon et al.	2018	Retrospective cohort	USA	An absolute 0.5 mg/dL increase in SCr or at least 50% increase in SCr from baseline	18.9%	Excluded	366	74/292 (25.3%)	8/74 (9.5%)	**0.008**
Tookhi et al.	2021	Retrospective cohort	SA	KDIGO	24.6%	Excluded	158	8/77 (10.3%)	17/81 (20.9%)	0.07
Rungkitwattanakul et al.	2021	Retrospective cohort	USA	KDIGO	N/A	Excluded	207	16/74 (21.6%)	5/67 (7.4%)	0.002
**Studies including non-critically ill patients only**										
Balcı et al.	2018	Retrospective cohort	Turkey	AKIN	NA	CKD: 14.4% AKI: 25%	132	26/63 (41.3%)	7/69 (10.1%)	**<0.001**
Robertson et al.	2018	Retrospective cohort	USA	An absolute 0.5 mg/dL increase in SCr or at least 50% increase in SCr from baseline	NA	Excluded	169	14/85 (16.5%)	3/84 (3.6%)	**0.009**
Mullins et al.	2018	Prospective cohort	USA	1.5-fold increase in SCr (baseline vs. within first 7 days of antimicrobial therapy)	NA	Excluded	143	28/94 (29.8%)	7/49 (14.3%)	**<0.001**
Rutter et al.	2019	Retrospective cohort	USA	RIFLE	NA	Excluded	10,236	2713/9898 (27.4%)	52/338 (15.4%)	**<0.001**
Ide et al.	2019	Retrospective cohort	Japan	KDIGO	NA	Not reported	76	9/27 (33.3%)	4/49 (8.2%)	**0.015 ***
**Studies including critically ill patients only**										
Schreier et al.	2019	Retrospective cohort	USA	AKIN	100%	CKD: 13.7% AKI: 29.3%	1926	601/1540 (39.0%)	135/386 (34.9%)	0.49
Blevins et al.	2019	Retrospective cohort	USA	KDIGO	100%	Excluded	758	144/366 (39.3%)	92/392 (23.5%)	**<0.0001**
Kang et al.	2019	Retrospective cohort	SK	KDIGO	100%	Excluded	157	39/74 (52.7%)	23/83 (27.7%)	**<0.0001**

Abbreviation: AKI: acute kidney injury; CKD: chronic kidney disease; V + PT: vancomycin/piperacillin-tazobactam; V + M: vancomycin/meropenem; SCr: serum creatinine; KDIGO: kidney disease improving global outcomes; RIFLE: risk, injury, failure, loss, and end-stage renal failure; AKIN: acute kidney injury network; SA: Saudi Arabia; SK: South Korea. * *p*-value for carbapenem group not meropenem only.

## Data Availability

Not applicable.

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
