# Peer review of "Comparative Risk of Acute Kidney Injury Following Concurrent Administration of Vancomycin with Piperacillin/Tazobactam or Meropenem: A Systematic Review and Meta-Analysis of Observational Studies"

_antibiotics, 2022, doi:10.3390/antibiotics11040526_

Round 1

Reviewer 1 Report

I would like to commend the authors for an excellently performed meta-analysis of the thorny issue regarding excess acute kidney injury (AKI) secondary to the association of vancomycin plus piperacillin (V+PT), and whether vancomycin plus meropenem (V+M) is a superior option. The research question is still under debate, but a multitude of recent studies do show an association between V+PT administration and AKI. At the same time, there appears to be no such association with other drug regimes such as vancomycin plus cefepime. The research presented argues for a lower incidence of AKI associated with V+M therapy, but cannot conclude that V+M is a superior option with regard to other measured outcome variables such as LOS, mortality and incidence of renal replacement therapy. The authors correctly identify the main caveat of this study: that the included study groups are highly heterogenous. I would argue that the lack of prospective data (1 single study) is also a weak point, especially from properly conducted blinded trials. Nevertheless, the research as presented is of interest to the field, and contributes to the growing body of evidence surrounding side-effects associated with V+PT versus V+M drug therapies. I have no major reservations with this work.

Minor comments:

  • The images included are of low resolution, and are heavy on tabular data. They appear to be direct output from the RevMan software. I would prefer the tabular data in the images to be presented as actual tables, and the odds ratio plots to be presented as images - perhaps as a multi-panel figure.
  • The main analysis includes one study (Tookhi et. al 2021) which has a strongly deviating odds ratio for AKI. The authors could discuss this study and the likely reason for the deviating result as compared to the other 11 studies. For example, the low study population count, low quality indicator for AKI (SCr), higher incidence of critical illness in the V+M groups.

Author Response

The authors of this manuscript would like to thank you for your valuable comments. Please see the attachment. 

Reviewer 2 Report

Thank you for submitting this good research.

 There's one thing I point out, in the section of results (3.3 outcomes from the subgroup analyses based on clinical setting). In terms of AKI, the following area : Only one study presented a significant increase in the odds of AKI among V+PT users versus V+M users in patients treated in ICU and non-ICU settings with aOR=7.03;95% CI 1.97-25.09; Figure 3) require correction to decide the exact location of parenthesis. 

Author Response

(The authors gave the same response as above.)

Reviewer 3 Report

The systematic review and meta-analysis compared the risks of AKI following antibiotic treatment of Vancomycin with Piperacillin/Tazobactam or Meropenem. The authors selected observational studies and evaluated the nephrotoxicity by analysing the AKI events, hospital length of stay, RRT and mortality. Overall the study has been well designated, however, some improvements are required. 

1)Appropriate list of the inclusion/exclusion criteria is necessary, including the characteristics of the patient (age, comorbidities, etc). 

2)Description of the methodology used for the quality and risk of bias assessment. It is mentioned that the authors used NOS, but they should justify the reason for their choice. 

3)Since they selected studies with different patient characteristics a table describing the study characteristics would help. 

4) Forest plots are too small and all need a better resolution.

5) Some outcomes have been evaluated in only a few studies, an explanation for this is necessary. 

6) A linguistic check and proofreading is required. 

Author Response

(The authors gave the same response as above.)

Round 2

Reviewer 3 Report

The revised form of the manuscript has been improved and I am happy with the amendments made by the author. Still a few typos present

This manuscript is a resubmission of an earlier submission. The following is a list of the peer review reports and author responses from that submission.